# Novel Small Molecules in IBD: Current State and Future Perspectives

**DOI:** 10.3390/cells12131730

**Published:** 2023-06-27

**Authors:** André Jefremow, Markus F. Neurath

**Affiliations:** 1Department of Medicine 1, University Erlangen-Nürnberg, 91054 Erlangen, Germany; markus.neurath@uk-erlangen.de; 2Deutsches Zentrum Immuntherapie (DZI), 91054 Erlangen, Germany; 3Comprehensive Cancer Center Erlangen-EMN (CCC ER-EMN), 91054 Erlangen, Germany

**Keywords:** inflammatory bowel diseases, Crohn’s, ulcerative colitis, small molecules, small molecular drugs, JAK

## Abstract

Biologicals have dominated the therapeutic scenery in inflammatory bowel diseases (IBDs), namely ulcerative colitis (UC) and Crohn’s disease (CD), for the past 20 years. The development of tofacitinib was the starting point for an era of small molecules after the era of biologicals. These new agents may challenge the use of biological agents in the future. They share properties that appeal to both patients and physicians. Low production costs, a lack of immunogenicity, and ease of use are only some of their benefits. On the other hand, patients and their physicians must manage the potential side effects of small molecules such as JAK inhibitors or S1P1R modulators. Here, we present agents that have already entered the clinical routine and those that are still being investigated in clinical trials.

## 1. Introduction

The development of the anti-tumor necrosis factor α (TNF-α) antibody infliximab defines a ground-breaking moment in the treatment of inflammatory bowel diseases (IBDs), namely Crohn’s disease (CD) and ulcerative colitis (UC). Its efficacy in the induction and maintenance of remission reduced symptom burdens in many IBD patients and enabled them to lead normal lives [1,2,3]. Above that, it represents the beginning of an “era of biologicals” in IBD treatment and has resulted in further developments, such as anti-interleukin (IL) 12/23 and anti-α4β7-integrin antibodies [4]. New treatment goals, including “deep remission” and “mucosal healing”, accompanied this progress [5,6,7].

Yet, a large proportion of patients suffer from primary or secondary treatment failure.

Up to 40% of IBD patients experience no improvement at all through anti-TNF therapy [8,9]. More than one-third of patients treated with anti-TNF antibodies lose their response over time; annually, almost 21% [10]. In general, many patients show at some point a loss of response or therapy failure after treatment with biologicals. Zhao and coworkers showed that only 44.3% of UC patients and 59.9% of patients with CD received their first biological treatment after one year. Three years after the beginning of therapy, treatment persistence decreased to 16.9% and 33.6%, respectively; 32.8% of all patients required a second-line treatment with a different biological agent [11,12].

Coincidently, therapy becomes more difficult after treatment failure: If patients received an anti-TNF antibody in the past, the anti-α4β7-integrin antibody vedolizumab and the sphingosine 1-phosphate receptor modulator ozanimod are less effective in comparison with treatment with these substances in anti-TNF-naïve patients [12]. Whether another anti-TNF antibody results in a sustained response depends largely on the reason that led to treatment discontinuation with the first anti-TNF antibody. If intolerance resulted in stopping therapy, 61% of the patients achieve remission when doctors treat them with a different anti-TNF agent. Patients with primary or secondary treatment failure of anti-TNF treatment have a benefit in only 30% and 45%, respectively, when they receive a second-line anti-TNF therapy. Primary non-responders are 24% more unlikely to show remission with a different biological in comparison with patients who stopped because of intolerance [8]. This underlines the desire for different therapy approaches in the treatment of IBD [13].

The terms “small molecules” and “small molecular drugs (SMDs)” describe a heterogeneous group of drugs that share common features; e.g., the ability to cross biological barriers, modulation of different biological targets, and oral bioavailability. Several characteristics define the “rule of five” that enables oral bioavailability [14,15].

Recent developments of new drugs, e.g., Janus kinase (JAK) inhibitors, seem to have started an era of small molecules in IBD treatment after antibody-based approaches dominated the landscape for almost two decades. Here, we show small-molecule drugs that have already entered clinical practice and those in advanced clinical studies (phase II and phase III trials).

This narrative review focuses on recent developments showing the different classes of small molecules that have made it into the clinical routine since biological agents have become a treatment standard. We also highlight substances that are still under investigation. Beyond that, we display some trials that failed. We conducted a MEDLINE search using the key words “small molecules” in combination with “IBD”, “inflammatory bowel diseases”, “Crohn’s disease”, and “ulcerative colitis” from January 2020 until April 2023. Above that, we added studies we found noteworthy.

## 2. JAK Inhibitors

Cytokines, for example IL-9, IL-23, and interferon (INF)-γ, depend on the Janus kinases (initially termed “Just Another Kinase”) for phosphorylation of signal transducer and activator (STAT) transcription factors, because their receptors lack this ability. JAK 1, JAK 2, JAK 3, and tyrosine kinase 2 (TYK2) constitute this enzyme family. After phosphorylation, STAT (seven family members: STAT1–4, 5A, 5B, and 6) proteins enter the nucleus and regulate gene transcription (Figure 1) [16,17,18]. Worthy of notice, Barrett and coworkers identified JAK-2-related gene loci in association with a risk of developing CD, while Anderson et al. revealed this for UC [19,20].

Important side effects of JAK inhibitors include infections and cardiovascular events. Infections of the upper airways, including influenza, are the most common ones.

### 2.1. Tofacitinib

Sandborn and his coinvestigators showed in a landmark study that tofacitinib, a non-selective JAK inhibitor, led to clinical responses in UC in a dose-dependent manner [21]. Later, Sandborn et al. carried out the OCTAVE trials. Here, they demonstrated the potential of tofacitinib to induce and maintain remission (defined as a total Mayo score of ≤2, with no subscore >1 and a rectal bleeding subscore of 0). While 18.5% of the patients receiving tofacitinib (10 mg BID) achieved remission, only 8.2% in the placebo arm met the primary endpoint. After 52 weeks, 34.3% (tofacitinib 5 mg BID), 40.6% (tofacitinib 10 mg BID), and 11.1% (placebo) reached remission. The authors detected mucosal healing in the treatment group significantly more often than in the placebo group as well. Infections (especially herpes zoster) and cardiovascular diseases occurred more often in the treatment arm than in the placebo arm [22]. About half of the patients had been pretreated with anti-TNF antibodies.

Crohn’s disease patients did not attain remission significantly more frequently with tofacitinib than with placebo [23] (Table 1).

### 2.2. Filgotinib

Feagan and colleagues conducted the SELECTION trial. Here, they validated the efficacy of the selective JAK 1 inhibitor filgotinib to achieve remission in UC and maintain it. Totals of 26.1% (induction study A) and 11.5% (induction study B) of the filgotinib (200 mg) treated patients showed clinical remissions; the proportions in the placebo arms were 15.3% and 4.2%, respectively. After 58 weeks, 37.2% of the filgotinib (200 mg) treated patients had clinical remission versus 11.2% in the placebo group. There was no statistical significance for achieving the primary endpoint after 10 weeks, when patients were treated with a dose of 100 mg, but it appeared after 58 weeks (23.8% vs. 13.5%). There was one patient suffering from pulmonary embolism in the filgotinib 200 mg arm and four total cases of herpes zoster in the filgotinib groups (one in the 100 mg group and three in the 200 mg group) [24].

The DIVERSITY trial failed to show the efficacy of filgotinib to induce and maintain remission in Crohn’s disease, as the sponsor noted in a press release.

### 2.3. Upadacitinib

Another JAK 1 inhibitor, upadacitinib, is approved not only for UC, but also for CD.

U-ACHIEVE induction (UC1) and U-ACCOMPLISH (UC2) investigated the ability of upadacitinib to induce remission, while its effect on the maintenance of remission was tested in U-ACHIEVE maintenance (UC3). Here, Danese et al. demonstrated that 26% (UC1) and 34% (UC2) of the patients went into clinical remission after 8 weeks. The corresponding proportions in the placebo arms were 5% and 4%, respectively. Of the patients, 42% who were treated with a dose of 15 mg upadacitinib, and 52% of those receiving a higher dose (30 mg), showed clinical remission after 52 weeks. At the same time, only 12% of the patients in the placebo arm performed comparably [25]. A phase II trial preceded these studies [31].

Sandborn et al. outlined the capability of upadacitinib to induce and perpetuate remission in CD. They treated patients with upadacitinib 3 mg BID, 6 mg BID, 12 mg BID, 24 mg BID, 24 mg once daily, or placebo. Of these patients, 13% (3 mg BID), 27% (6 mg BID), 11% (12 mg BID), 22% (24 mg BID), and 11% (24 mg once daily) achieved clinical remission after 16 weeks in comparison with 11% in the placebo arm. Additionally, 10% (3 mg BID), 8% (6 mg BID), 8% (12 mg BID), 22% (24 mg BID), and 14% (24 mg once daily) achieved endoscopic remission after 16 weeks in comparison with no patient in the placebo arm. Adverse events comprised infections, including serious infections, and an increase in high- and low-density lipoprotein cholesterol [32]. In the U-EXCEL and U-EXCEED phase III trials, 49.5% and 38.9% of the patients treated with 45 mg of upadacitinib showed clinical remission, respectively, in comparison with 829.1% and 21.1% of the patients from the placebo group. U-ENDURE investigated upadacitinib in maintenance therapy. Here, the remission rates were 37.3% (15 mg upadacitinib), 47.6% (30 mg), and 15.1%. Furthermore, 27.6% (15 mg), 40.1% (30 mg), and 7.3% (placebo) of the patients experienced an endoscopic response. [26].

Noteworthy, JAK inhibitors elevate the risk for developing herpes zoster infections. While we know from many rheumatologic trials that JAK inhibitors increase the risk for major adverse cardiovascular events (MACEs) in patients with rheumatoid arthritis (RA), post hoc analyses have confirmed that this applies to IBD patients, too.

Both infections and MACEs happen more frequently in the induction phase than during maintenance therapy, showing a dose dependence. Physicians should consider a patient’s individual risk for adverse events [26].

To sum it up, JAK inhibitors show a great potential in inducing and maintain remission in IBD patients. Yet, there is no consensus on when to use them (e.g., first-line treatment or after one or several biologicals). In our experience, upadacitinib has the greatest impact and the greatest risk of adverse events. Its use in difficult-to-treat UC and Crohn’s in otherwise healthy patients (with no cardiovascular risk factors) seems appropriate. Any JAK inhibitor can be used in UC. Whether physicians use a JAK inhibitor or an antibody-based regime depends on the patient’s preference, comorbidities, and prior treatments.

### 2.4. TD-1473

The agent TD-1473 belongs to the group of pan-JAK inhibitors. Sandborn and his team investigated it in a phase Ib trial for patients with moderate to severe UC. Because only 40 patients participated, there was no statistical analysis. Yet, the researchers observed clinical response, endoscopic improvement, and a decrease in C-reactive protein (CRP) and fecal calprotectin [33]. Further studies to determine the agent’s impact on UC and CD activity have been terminated early.

### 2.5. Brepocitinib

Brepocitinib (PF-06700841), an inhibitor of JAK 1 and TYK2, and the JAK 3 inhibitor PF-06651600, are being studied in a phase IIa study for moderate to severe CD (NCT03395184) and in a phase IIb trial for moderate to severe UC (NCT02958865).

Trials to investigate another TYK2 inhibitor named deucravacitinib (BMS-986165) in moderate to severe UC (NCT03934216) and CD (NCT03599622) are running.

## 3. Cobitolimod

Toll-like receptors (TLRs) play a crucial role in the innate immune system. Modulating them seems a promising approach to treat autoimmune disorders. Atreya and his team investigated cobitolimod, a DNA-based immunomodulatory sequence serving as a TLR-9 receptor agonist, in moderate to severe UC. TLR-9 activation leads to the induction of anti-inflammatory cytokines such as IL-10 and type I interferons (Figure 2). As a topical agent, cobitolimod was administered during endoscopy. Even though the study did not reach its primary endpoint (clinical remission), it met secondary endpoints (e.g., relief of symptoms, mucosal healing, and histologic improvement) [34].

The CONDUCT trial was limited to left-sided ulcerative treatment because administration of the study drug was performed via enema. Here, 250 mg of topical cobitolimod administered twice induced clinical remission significantly more often than placebo (21% vs. 7%) [35]. A phase III trial is recruiting patients now (NCT04985968).

## 4. Anti-Leukocyte Trafficking Agents

Anti-integrin agents were shown to be effective in the treatment of IBD. Vedolizumab, an anti-α4β7-integrin antibody, prevents gut homing and thereby induces and maintains remission. In contrast to the anti-α4-integrin antibody natalizumab, vedolizumab contains no risk of progressive multifocal leukoencephalopathy (PML) due to JC virus reactivation.

### 4.1. Carotegrast (AJM300)

AJM300 targets α4-integrin similarly to natalizumab, though it is taken orally. Sugiura and his team tested it in an experimental colitis model with mice lacking IL-10 in CD4+ t cells [36]. Yoshimura et al. conducted a phase IIa trial. Patients receiving AJM300 in a dose of 960 mg three times a day showed a clinical response rate of 62.7% in comparison with 25.5% of the placebo group. Clinical remission appeared in 23.5% and 3.9% in the AJM300 and placebo groups, respectively. The investigators did not see serious side effects, but larger sample sizes are needed to confirm this, especially with respect to PML. Yet, they discovered nasopharyngitis as an adverse event and observed an increase in the peripheral leukocyte count, possibly as a result of integrin blockade [17,27]. Matsuoka and his team conducted a phase III study in patients with moderate UC, defined as a Mayo clinical score of 6–10. Here, 45% of the patients in the AJM300 group and 21 (21%) patients in the placebo group showed a clinical response after 8 weeks. Again, nasopharyngitis was the most observed adverse event and was usually mild to moderate [37].

### 4.2. Alicaforsen

Blocking intercellular adhesion molecule 1 (ICAM-1) with alicaforsen, an ICAM-1 antisense oligonucleotide, failed to reach the trials’ primary endpoints for treatment of mild to moderate UC [28,38], even though it had showed promising results before [39,40].

## 5. Sphingosine-1-Phosphate Receptor Modulators

Sphingosine-1-phosphate receptor modulators act in a manner comparable to anti-leukocyte trafficking agents. Sphigosine-1-phosphate acts as a ligand on the sphingolipid ligand of G-protein-coupled receptors (S1P1–S1P5). These receptors orchestrate the discharge of leukocytes out of the lymphoid tissues (lymph nodes). By modulating S1P receptors, fewer pro-inflammatory leukocytes enter peripheral tissues. Their clinical use has been established in multiple sclerosis.

### 5.1. Ozanimod

Ozanimod reveals a specific modulation of S1P1 and S1P5. The TOUCHSTONE trial met its primary endpoint of clinical remission (Mayo clinic score ≤2; no subscore >1) in UC at week 8 in 16% of the ozanimod 1 mg group, 14% of the ozanimod 0.5 mg group, and 6% of the placebo group. Clinical remission was maintained in 21% of the 1 mg group, 26% of the 0.5 mg group, and only 6% of the placebo group at week 32. Bradycardia and elevated liver enzymes appeared in the verum groups [41].

Sandborn and coworkers conducted the True North trial as a phase III study. Here, they investigated the efficacy of ozanimod with respect to the induction and maintenance of clinical remission (three-component Mayo score). While ozanimod could achieve induction in 18.4% of the patients, only 6.0% of patients reached this goal in the placebo arm. Of the ozanimod-treated patients, 37.0% maintained in remission, but only 18.5% of the placebo-receiving patients did so. In this trial, elevated liver enzymes and bradycardia represented serious adverse events, while the rate of infection was similar in the treatment and placebo arms [42].

Feagan et al. studied ozanimod in patients with moderate to severe CD. The STEPSTONE trial showed that 23.2% of the patients had an endoscopic response defined as a reduction in the simple endoscopic score for Crohn’s disease (SES-CD) at 12 weeks in comparison with baseline. Clinical remission (defined as CDAI <150) appeared in 39.1% of patients. Patients received ozanimod in a 7-day dose escalation (4 days on ozanimod 0.25 mg daily followed by 3 days at 0.5 mg daily). After that, ozanimod 1.0 mg was given until week 12. There was no control arm. Phase III trials for ozanimod in CD therapy are recruiting patients (NCT03440372, NCT03440385, NCT03464097, NCT03467958) [17,43].

### 5.2. Etrasimod

A different S1P modulator called etrasimod exhibits selectivity for S1PR1, S1PR4, and S1PR5. Sandborn and his team ran a study in which they compared the effect of 2 mg etrasimod, 1 mg etrasimod, and placebo in moderate to severe UC. They defined a primary endpoint as an improvement in the modified Mayo score. Patients receiving the 2 mg dose met this endpoint, while there was no significance for 1 mg in comparison with placebo. Above that, patients with 2 mg etrasimod achieved secondary endpoints (41.8 vs. 17.8), improvement in the total Mayo score in the two-component Mayo, as well as histologic remission (Geboes score <2) more often than patients from the placebo group.

Noteworthy, 7 of 102 patients from the etrasimod group stopped the treatment due to adverse events, while none from the placebo group did. Upper respiratory tract infections, nasopharyngitis, and anemia were observed mostly.

One patient experienced atrioventricular block second degree (type 1) and heart rate lowering considered to be treatment-related. Another patient experienced one event each of atrioventricular block first degree and sinus bradycardia without treatment relation. Both patients received 2 mg of etrasimod [29].

Based on these findings, Sandborn and coworkers conducted the ELEVATE trials, two randomized, double-blind, placebo-controlled, phase III studies. Here, 2 mg of etrasimod was compared with placebo in patients with moderate to severe UC with regard to clinical remission. The primary endpoint was clinical remission as a combined stool frequency subscore = 0 (or stool frequency subscore = 1 with a ≥1-point decrease from baseline), rectal bleeding subscore = 0, and endoscopic subscore of 1 or less. After 12 weeks, 27% of the treated patients went into clinical remission, compared with 7% of the placebo group in ELEVATE UC 52. Furthermore, 32% of the treated patients achieved this goal after 52 weeks, while 7% of the patients in the placebo arm experienced clinical remission. In ELEVATE UC 12, 25% of the etrasimod-treated patients showed clinical remission, while only 15% of patients receiving placebo reached this goal after 12 weeks. After 52 weeks of treatment, all secondary endpoints such as sustained clinical remission and corticosteroid-free remission with no use of corticosteroids for at least 12 weeks were met in the verum group [30]. Further phase III trials investigating etrasimod in UC are recruiting patients currently (NCT03950232, NCT04176588). Above that, patients with CD can participate in a phase II/III trial using etrasimod (NCT0417327).

### 5.3. Amiselimod

D’Haens et al. investigated the selective S1P1 modulator amiselimod with respect to CD. Unfortunately, no significant portion of patients showed a 100-point decrease in the Crohn’s disease activity index (CDAI), which was the primary endpoint. A total of 54.1% of the placebo group achieved this goal, unlike 48.7% of the amiselimod group. The authors could not detect a difference regarding previous exposure to anti-TNF agents or concomitant use of oral corticosteroids [44].

### 5.4. Phosphodiesterase 4 Inhibitors

Phosphodiesterases (PDE1–PDE11) catalyze the breakdown of cyclic adenosine monophosphate (cAMP) and cyclic guanosine monophosphate (cGMP). The activation of nuclear transcription factor kappaB (NF-κB) follows this event in different sets of immune cells, resulting in proinflammatory signaling. Hence, the use of phosphodiesterase 4 inhibitors in IBD could improve the inflammation status.

### 5.5. Apremilast

Danese et al. conducted a phase II study investigating the phosphodiesterase 4 inhibitor agent apremilast in active UC. Even though the primary endpoint of clinical remission (Mayo score ≤2; no individual score >1) was not met, the results seem promising: 31.6% in the 30 mg group, 21.8% in the 40 mg group, and 12.1% in the placebo group achieved clinical remission. Even 73.7% of the patients treated with 30 mg apremilast showed endoscopic response (40 mg: 21.8%; placebo: 12.1%). A total of 25.5% in the 40 mg group and 21.1% in the 30 mg group experienced headaches (placebo: 6.9%) [45].

### 5.6. Obefazimod

Obefazimod (ABX464) is a unique agent that was initially developed for treatment of human immunodeficiency virus (HIV) [46]. It interacts with cap-binding complexes (CBC) 20 and 80. Thus, it stabilizes the connection of these complexes, which ultimately leads to an enhanced production of micro-RNA (Figure 3). Chebli and colleagues found that treatment with obefazimod attenuated in a murine colitis model. In this model, the production of IL-22 by activated macrophages played an important role [47]. Moreover, obefazimod treatment resulted in an increased expression of microRNA (miR)-124 [48]. Koukos et al. demonstrated that miR-124 reduces STAT3 phosphorylation and is down-regulated in colon specimens of pediatric UC patients [49].

Of patients in the obefazimod group, 35.0% and 70.0% achieved clinical remission and clinical response, respectively, in a phase IIa trial versus 11.1% and 33.3% in the placebo group. The endoscopic improvement was 50% and 11% [50].

A phase IIb trial showed that significantly more patients achieved clinical and endoscopic remission when treated with obefazimod in comparison with placebo. Above that, about 25% of the patients were refractory to biologicals and JAK inhibitors [51]. The fact that 90% of the patients had been treated with at least two different therapies, including anti-TNF agents, and that some sufferd from ulcerative proctitis (which is said to be a hard-to-treat form of UC) gained much attention [52].

Several phase II and III trials are recruiting UC patients currently (NCT05177835, NCT05507203, NCT05507216, NCT05535946) [53].

## 6. Conclusions

Small molecules enrich the therapeutic landmark enormously in IBD. Table 1 provides an overview of agents that have been approved or have completed phase II trials. These agents show attractive features to both patients and physicians. Patients might favor the route of administration (orally or topically) over antibodies that need injections or infusions. The ability of small molecules to overcome biological barriers offers new mechanisms physicians can affect. While antibodies attach to cells and cytokines, small molecules develop effects within the cells, e.g., by inhibiting or promoting transcription. Additional attributes, such as a fast onset of action and a high impact in pretreated patients, impress patients as well as physicians. The short half-life of SMDs makes it easier to manage adverse events, for example, infections. Another key advantage in comparison with biologicals lies within their low risk of immunogenicity. Furthermore, production seems to be cheaper and easier than the manufacture of biologicals [54]. Yet, both patients and physicians need to learn how to handle the new side effects that may limit the usage of small molecules. Even though emerging therapy concepts improve patient care, we are still far from healing, underlining the urgent need for further research.

## Figures and Tables

**Figure 1 cells-12-01730-f001:**
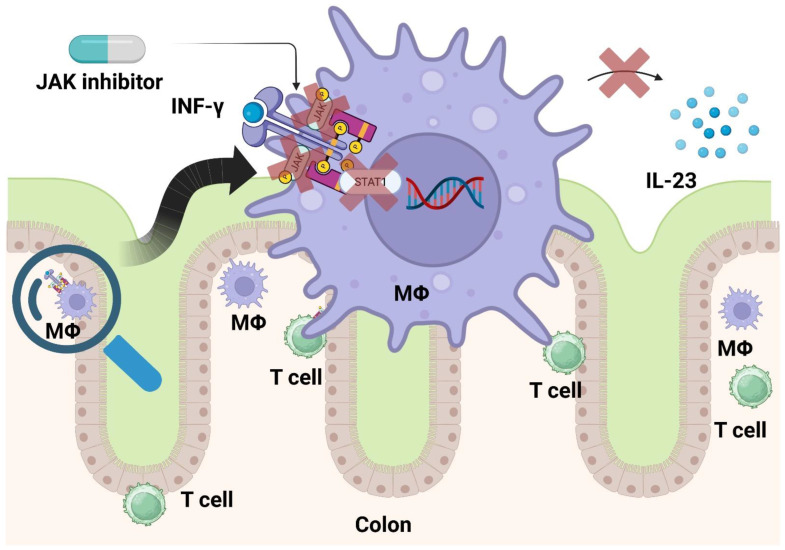
When cytokines (e.g., INF-γ) activate immune cells such as macrophages, cytokine receptors depend on JAK to activate the transcription factors of the STAT family. STAT promotes pro-inflammatory gene expression, resulting in, e.g., release of IL-23. JAK inhibitors prevent cytokine signaling at the beginning of the intracellular cascade. Created with biorender.com.

**Figure 2 cells-12-01730-f002:**
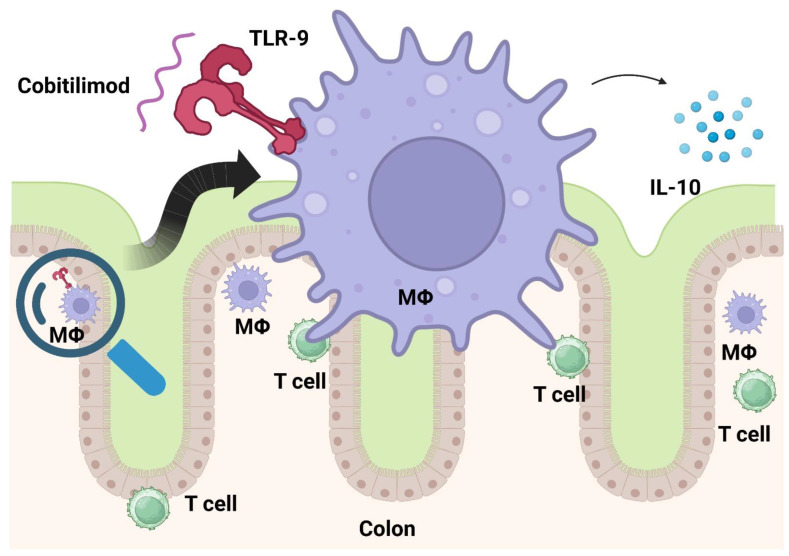
Cobitolimod activates as an oligodeoxynucleotide TLR-9 on immune cells and thereby releases anti-inflammatory cytokines (e.g., IL-10). Created with biorender.com.

**Figure 3 cells-12-01730-f003:**
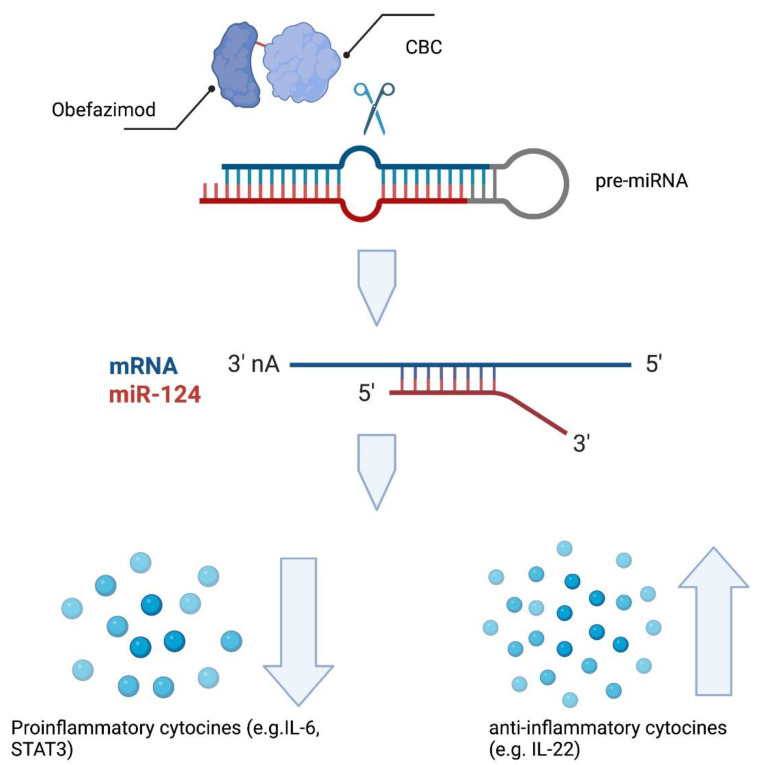
Obefazimod binds to and stabilizes cap-binding complex (CBC), leading to enhanced splicing of the miR-124 precursor, and thus, increased production of miR-124. This results in less pro-inflammatory signaling (e.g., via IL-6 and STAT3). Created with biorender.com.

**Table 1 cells-12-01730-t001:** Percentage of patients who achieved and maintained remission in trials in comparison with placebo. Highlighted numbers derive from phase III trials leading to approval.

Crohn’s Disease			Ulcerative Colitis		
Small Molecules	Induction	Maintenance	Induction	Maintenance	References
Tofacitinib ^1^			**18.5 vs. 8.2**	**34.3 vs. 11.1**	[22]
Filgotnibib			**26.1 vs. 15.3**	**37.2 vs. 11.2**	[24]
Upadacitinib ^2^	**49.5 vs. 29.1**	**47.6 vs. 15.1**	**34 vs. 4**	**52 vs. 12**	[25,26]
Cobitilimod			21 vs. 7		[27]
Carotegrast			45 vs. 21		[28]
Ozanimod			**18.4 vs. 6.0**	**37.0 vs. 18.5**	[29]
Etrasimod			25 vs. 15	32 vs. 7	[30]

^1^ Tofactinib: induction 10 mg BID, maintenance 5 mg BID; ^2^ Upadacitinib: induction 45 mg, maintenance 30 mg.

## Data Availability

No new data were created or analyzed in this study. Data sharing is not applicable to this article.

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
