# Peer review of "Novel Small Molecules in IBD: Current State and Future Perspectives"

_cells, 2023, doi:10.3390/cells12131730_

Round 1
Reviewer 1 Report
Thank you for this exhaustive description of the small molecules currently available and (for some) possibly available in the near future in IBD.
Few comments:
- there are few data concerning the cardiovascular and infectious side effects of JAKi. It may be interesting to emphasize this point and to underline the weight of the dose used. And also to make some remarks on how to chose in clinical practice between the available JAKi
- data from DIVERSITY are now available and can be included
- there is a possible problem with the classification of the molecules (as ozanimod/etrasimod are S1P receptor modulator and apremilast is a phosphodiesterase inhibitor). Can you check the numbering of the paragraphs?
- reference 29 does not seem to have a correct formatting
Author Response
Dear reviewer 1,
please see the uploaded file. Thank you very much!

Reviewer 2 Report
The aim of the current study was to give an overview on the current and future perspectives of small molecules in IBD. The topic is important, relevant and rapidly evolving, however, the review is not systematic, has no real methodology and suffers from inconsistencies and typos. It is not clear how the literature search was performed, what are the exact aims and how the specific small molecules were chosen to be presented. Older molecules, such as 5-ASA, thiopurines, methotrexate and tacrolimus are not mentioned. This is OK, but in this case, the title should probably be changed to "Novel small molecules….". Further, it is not clear why the authors chose to discuss some small molecules that have been shown to be ineffective, such as Alicaforsen, Amiselimod or Apremilast. Again, this is OK, but in this case, why not include other small molecules that failed to reach the market such as Mongersen.
For ease of the readers, I would suggest adding a table that summarizes all the therapies and their efficacies.
Further comments-
1. In the JAK inhibitor section- why is Filgotibinb- that is not yet approved, presented before upadacitinib?
2. Lines 120-130- Citing the results of the phase 2 trials seems less relevant as the results of the Phase 3 (Excel, Exceed, Endure) trials, that are cited here, have been presented at the ECCO and DDW meetings. The FDA has recently approved Upa for CD based on those results. It would probably be more relevant to cite the figures from those phase 3 trials which use the currently approved dosages of Upa for CD, instead of the Phase 2 dosages and results.
3. Line 115- consider adding here and in all other relevant trials (perhaps in a table) the delta between percent achieving clinical remission in the intervention compared to the placebo groups.
4. Line 117- "maintained clinical remission…"- the word maintained is incorrect given that not all patients entering the maintenance phase were in clinical remission.
5. Line 168-170- I believe that the mice studies are not relevant to the current review.
6. Line 181-184- why relevant?
7. Line 200- in most instances the authors cited the names of the trials. Here (the True North trial) it is not cited from an unclear reason.
8. Line 256- "3.7. Phosphodiesterase 4 Inhibitors". If the authors do choose to discuss molecules that have failed to be approved for IBD, they should probably put these medications in a category by itself and not as part of the leukocyte trafficking inhibitors.
9. Obefazimod should also be in a different category then leukocyte trafficking inhibitors.
1. Line 25, 84 and any multiple other lines– typo "und"
2. Line 102- not clear. The word "but" should probably be replaced.
3. Line 116- should be 52%
4. Line 141- typo "fur"
5. Line 145- typo
6. Line 150- the word "it" should be erased.
7. Line 155- "twice" daily?
8. Line 178- " Here, 45%) patients"
9. Many additional typos should be corrected. English is good, although the use of the work "like" should be probably be diminished and replaced.
10. Line 270, 271, 275, 293- typo- should be "obefazimod"
Author Response
Dear reviewer 2,
please see the uploaded file. Thank you very much!

Round 2
Reviewer 2 Report
My comments have been addressed.
Still some minor typos and grammar issues: line 45, line 68, lines 85, 153, 277